# Degradation of Four Major Mycotoxins by Eight Manganese Peroxidases in Presence of a Dicarboxylic Acid

**DOI:** 10.3390/toxins11100566

**Published:** 2019-09-27

**Authors:** Xiaolu Wang, Xing Qin, Zhenzhen Hao, Huiying Luo, Bin Yao, Xiaoyun Su

**Affiliations:** 1Key Laboratory for Feed Biotechnology of the Ministry of Agriculture, Feed Research Institute, Chinese Academy of Agricultural Sciences, Beijing 100081, China; xiaolu4444@126.com (X.W.); xingqin@hust.edu.cn (X.Q.); zhenzhenhao2012@163.com (Z.H.); luohuiying@caas.cn (H.L.); 2College of Life Science and Technology, Huazhong University of Science and Technology, Wuhan 430074, China

**Keywords:** manganese peroxidase, mycotoxin, aflatoxin, zearalenone, deoxynivalenol, fumonisin, detoxification

## Abstract

Enzymatic treatment is an attractive method for mycotoxin detoxification, which ideally prefers the use of one or a few enzymes. However, this is challenged by the diverse structures and co-contamination of multiple mycotoxins in food and feed. Lignin-degrading fungi have been discovered to detoxify organics including mycotoxins. Manganese peroxidase (MnP) is a major enzyme responsible for lignin oxidative depolymerization in such fungi. Here, we demonstrate that eight MnPs from different lignocellulose-degrading fungi (five from *Irpex lacteus*, one from *Phanerochaete chrysosporium*, one from *Ceriporiopsis subvermispora*, and another from *Nematoloma frowardii*) could all degrade four major mycotoxins (aflatoxin B_1_, AFB_1_; zearalenone, ZEN; deoxynivalenol, DON; fumonisin B_1_, FB_1_) only in the presence of a dicarboxylic acid malonate, in which free radicals play an important role. The *I. lacteus* and *C. subvermispora* MnPs behaved similarly in mycotoxins transformation, outperforming the *P. chrysosporium* and *N. frowardii* MnPs. The large evolutionary diversity of these MnPs suggests that mycotoxin degradation tends to be a common feature shared by MnPs. MnP can, therefore, serve as a candidate enzyme for the degradation of multiple mycotoxins in food and feed if careful surveillance of the residual toxicity of degradation products is properly carried out.

## 1. Introduction

Mycotoxins are a diverse, large group of fungal secondary metabolites that exert severe toxic effects on human and animals worldwide. A recent study revealed that, among the nearly 20,000 feed and feed raw material samples which were collected from Asia, Europe, Americas, Africa, and Middle East, 72% contained at least one of the five major mycotoxins (aflatoxin B_1_, AFB_1_; zearalenone, ZEN; deoxynivalenol, DON; fumonisin B_1_, FB_1_; and ocharatoxin A, OTA). See Figure 1 for structures [1]. In the United States, an annual loss of $52.1 million to $1.68 billion is estimated to be caused by merely aflatoxin contamination in the corn industry [2].

To minimize mycotoxin contamination, agronomic means have proven to be effective in reducing the risk but are still inadequate [1]. Physical and chemical treatments of food and feed do not remove mycotoxins effectively because these secondary metabolites are often insensitive. In feed industry, inert adsorbents such as montmorillonite are widely used to bind mycotoxins in the animal’s gastrointestinal tract. However, binding of mycotoxins to adsorbents is effective for only limited kinds of mycotoxins such as AFB_1_ because of the inherent selective nature of adsorbents [3]. Therefore, microbial and, in essence, enzymatic detoxification are increasingly regarded as an attractive method to control mycotoxin contamination. However, mycotoxins are structurally distinct to each other and, more importantly, different types of mycotoxins usually co-exist in the food, feed, and feed raw materials. For example, more than 38% of the samples were detected with co-contamination of two or even more mycotoxins [1]. To deal with the complexity of mycotoxin contamination, a feasible route toward economic detoxification would ideally be to use one or a few enzymes. This requires that enzymes must have a wide substrate specificity on multiple mycotoxins.

Most of the identified enzymes that are able to detoxify mycotoxins can be grouped into hydrolase, transferase, epimerase, and oxidoreductase. Although most discovered enzymes, if not all, are reported to detoxify only a certain type of mycotoxins, among the oxidoreductases, a special group of enzymes targeting lignin (i.e., laccase and lignin-modifying peroxidases) have intriguing promiscuous substrate specificity. It is thus noted that the lignin-degrading white rot fungi have been successfully used in mycotoxin detoxification [4,5]. In addition, part of mycotoxins may be structurally similar to a lignin monomer or its derivatives. For example, AFB_1_ has a coumarin structure (Figure 1), which is a derivative of the lignin monomer *p*-coumaryl alcohol [6]. Indeed, laccase (a multicopper oxidase catalyzing lignin degradation or polymerization) [7] and manganese peroxidase (MnP, a Mn^2+^-dependent lignin-degrading peroxidase) [8] are occasionally found to be able to degrade AFB_1_ [9,10,11]. MnP has a higher redox potential than laccase, suggestive of its higher ability to react with more recalcitrant substrates. MnP is widely distributed in lignin-degrading white-rot filamentous fungi. However, despite the previous finding of MnP to degrade AFB_1_, it remains unknown if MnP can serve as a candidate enzyme to degrade multiple mycotoxins. In this study, we determined the ability of MnP to act on other mycotoxins and if the ability to degrade multiple mycotoxins is a feature shared by MnPs. The recombinantly produced MnPs used in this study were from *Irpex lacteus* CD2, *Phanerochaete chrysosporium*, and *Ceriporiopsis subvermispora*. All these fungi have not been reported to have mycotoxin-degrading ability. However, they all express MnPs involved in lignin degradation. *I. lacteus* CD2 encodes seven MnPs [12], with two of them having the ability to oxidize the recalcitrant non-phenolic lignin model compound veratryl alcohol in the presence of malonate [13]. *P. chrysosporium* and *C. subvermispora* are the representative simultaneous and selective white rot lignocellulose degraders, respectively [14], both of which also encode MnP enzymes [15,16]. Therefore, five MnPs from *I. lacteus* CD2 (which can be recombinantly produced) and one each representative MnP from *P. chrysosporium* and *C. subvermispora* were tested for their ability to degrade multiple mycotoxins.

## 2. Results

### 2.1. Degradation of AFB_1_ and ZEN by IlMnP5 and IlMnP6

MnPs, frequently discovered in white rots, have a higher redox potential (up to 1.5 V) than laccase (up to 790 mV) [17,18]. Moreover, certain carboxylates, such as malonate, can further boost the ability of MnPs to oxidize chemicals with even higher redox potentials or more recalcitrant structures [13]. In addition to AFB_1_, ZEN is also a common mycotoxin contaminant of food or feed. Therefore, we started to investigate if MnP can degrade multiple mycotoxins by using ZEN in addition to AFB_1_ as the two most common mycotoxins in such a malonate buffer.

*I. lacteus* is known to encode seven MnP enzymes, among which *Il*MnP5 and *Il*MnP6 are the easiest to produce recombinantly. Therefore, *Il*MnP5 and *Il*MnP6, the two *I. lacteus* MnPs recombinantly produced in *E. coli*, were first tested for their ability to degrade AFB_1_ and ZEN, the two major mycotoxins with much differing chemical structures (Figure 1). The activities of the two enzymes were calibrated using 2,2′-Azino-*bis*(3-ethylbenzothiazoline-6-sulphonic acid (ABTS) as the substrate. The enzymes (0.5 U/mL each) were individually incubated with 5 μg/mL each of AFB_1_ or ZEN in 70 mM malonate buffer supplemented with 1 mM MnSO_4_ and 0.1 mM H_2_O_2_. The reaction was carried out at 30 °C for 9 h. Periodically, samples were taken out and three volumes of DMSO were added to terminate the reaction. HPLC was used to analyze the reaction products. It was thus discovered that degradation of AFB_1_ was fast for both enzymes (Figure 2a). The degrading percentages were 50.7% and 49.3% after 30 min, increased steadily to 84.9% and 83.7% at the end of 3 h, and then gradually ascended to 94.6% and 94.5% for *Il*MnP5 and *Il*MnP6, respectively, after 9 h. The degradation of ZEN was slower, with degradation percentages of 34.0% and 33.2% for *Il*MnP5 and *Il*MnP6, respectively, at 9 h (Figure 2b).

To have a first glimpse if AFB_1_ degradation may lead to detoxification, growth of hydra was inspected in a culture containing AFB_1_ treated or non-treated with one of the MnPs. Hydra has been used to determine the residual toxicity of AFB_1_ treated with ozone [19]. The hydra shrunk at 20 h and finally collapsed (40 h) when incubated with non-treated AFB_1_ (Figure 2c). In contrast, they remained alive even after 40 h of incubation in either of the MnPs-treated AFB_1_ samples. For ZEN, an assay using an engineered *S. cerevisiae* strain BLYES was employed. With an estrogenic activity, ZEN can bind to an intracellular estrogen receptor and stimulate emission of bioluminescence in this yeast (Appendix A) [20,21]. BLYES was thus grown to an optical density of 0.6 and incubated for 6 h with MnPs-treated ZEN. As a control, non-treated ZEN was diluted to a concentration at the mid-log phase of the responsive curve (Appendix A), allowing the most sensitive detection of ZEN degradation. Although only 33.2% of ZEN was degraded after 9 h of incubation, at the same dilution fold, the bioluminescence in the MnP-treated ZEN samples were significantly decreased to 10.6% (*Il*MnP5) and 14.2% (*Il*MnP6) (Figure 2d), which is consistent with the responsive curve. This indicated that enzymatic treatment by both MnPs largely mitigated the estrogenicity of ZEN (Figure 2d). From these experiments, a preliminary impression was deduced that the action of *Il*MnP5 and *Il*MnP6 may lead to detoxification of the AFB_1_ and ZEN mycotoxins with these model systems.

LC-MS/MS analyses of AFB_1_ transformation by *Il*MnP5 and *Il*MnP6 (Appendix A) revealed similar reaction as reported by other researchers [11].

A [M+H]^+^ ion was found at *m*/*z* 329.1888 (Appendix A). Its molecular weight was 16 Da higher than AFB_1_, suggesting that 8,9-vinyl bond of AFB_1_ might be oxidized by MnP and an oxygen atom was added to the molecule to form AFB_1_-8,9-epoxide (Figure 3a). Note that the product ions *m*/*z* 313.0744, 285.0705, 268.9351 were exactly the same with AFB_1_ standard, indicative of same mother nucleus structures. For ZEN transformation, a [M+H]^+^ ion was found at *m*/*z* 283.2819 (Appendix A). This indicated that the removal of the two hydroxyl group might occur in the benzene ring of ZEN (Figure 3b).

### 2.2. Radicals Play an Important Role in Mycotoxin Degradation by IlMnP5 and IlMnP6

The success of degrading two structurally different mycotoxins by *Il*MnP5 and *Il*MnP6 was clearly indicative of substrate promiscuity. In MnP-participated lignin degradation, the nonspecific reaction is underscored by the generation of free radicals, which can attack diverse linkages in the lignin polymer. We next determined if free radicals were also involved in the detoxification of AFB_1_ and ZEN. Transformation of AFB_1_ depended heavily on the buffer system because changing malonate to either acetate or lactate almost completely abolished the degrading ability of *Il*MnP5 and *Il*MnP6 (Figure 4a). In contrast, both enzymes could transform ZEN in the presence of malonate, acetate, and lactate (Figure 4b). Only very minor product peaks were observed when AFB_1_ reacted with MnPs in malonate. Two small peaks appeared in MnP-treated AFB_1_ samples, but no other new peaks were observed even when the products were extensively scanned from 190 to 400 nm (data not shown). Similar to the situation in AFB_1_ degradation, only very small product peaks were observed for the reactions with ZEN in presence of malonate buffer where radicals seemed to play a vital role. No other new peaks were observed when the products were also scanned from 190 to 400 nm for ZEN treated with *Il*MnP5 and *Il*MnP6 in presence of malonate (data not shown). In acetate buffer where free radicals could not be generated, a large product peak (with retention time: 20.3 min) was observed. The α-hydroxyl carboxylate lactate could serve as a weaker chelating reagent with Mn^3+^ [22], thus also leading to free radical generation. 

Mn^3+^, an oxidized product by MnP, chelates with malonate and leads to the release of alkyl, alkylperoxy, and superoxide radicals [13,23]. Free radicals can be eliminated by treatment using an enzyme or chemical scavenger. Superoxide dismutase (SOD) is effective in eliminating superoxide radical, while rutin is a strong alkylperoxyl radical scavenger [24]. We found that, in malonate buffer, SOD slightly but significantly, while rutin largely inhibited the degradation of AFB_1_ and ZEN, by *Il*MnP5 and *Il*MnP6 (Figure 4c–f). These data undoubtedly demonstrated that radicals are indeed involved in MnP-catalyzed mycotoxin degradation and therefore are responsible for the nonspecific reactions. In malonate, with part of ZEN being degraded, small product peaks were again observed. Acetate is not able to chelate Mn^3+^. However, a large product peak (retention time: 20.3 min) was observed for ZEN reacted with MnPs in this buffer (Figure 4b). The significant ZEN transformation in acetate suggests that *Il*MnP5 and *Il*MnP6 directly reacted with ZEN, possibly by oxidizing the phenolic hydroxyl groups, which is a typical feature of MnPs [22]. Lactate is an α-hydroxyl carboxylate and can also chelate with Mn^3+^ [22]. However, a significant product peak of 20.3 min was also observed when ZEN was incubated with MnPs in lactate (Figure 4b).

### 2.3. Degradation of Multiple Mycotoxins Is a common Feature Shared by Manganese Peroxidases

We then proceeded to investigate if *Il*MnP5 and *Il*MnP6 could also degrade other mycotoxins and, more importantly, if the ability to degrade multiple mycotoxins is a feature shared by other MnPs, as well. For this purpose, the other three main mycotoxins as addressed above, i.e., DON (10 μg/mL), FB_1_ (10 μg/mL), and OTA (50 μg/mL), were selected for testing. DON and FB_1_ do not have a phenolic hydroxyl group. Therefore, although for ZEN degradation, MnP performed best in the acetate buffer, to maximize the substrate promiscuity of MnPs, the malonate buffer (but not the acetate buffer) favoring radical production was used.

Eight MnP genes, including *IlMnP5* and *IlMnP6* and the other three from *I. lacteus* (*IlMnP1*, *IlMnP2*, and *IlMnP4*, *IlMnP5*, and *IlMnP6*) [12], one from *P. chrysosporium* (*PcMnP1*) [11,25], one from *C. subvermispora* (*CsMnP*) [26], and one from *N. frowardii* (*NfMnP*) were either expressed as recombinant proteins or purchased (for *Nf*MnP, from Sigma-Aldrich, catalog # 41563), were selected for testing (Appendix A). In the 72-h incubation when the reactions commonly reached a plateau (Appendix A), although the eight MnP enzymes displayed only negligible activity on OTA, they were quite effective in degrading AFB_1_, ZEN, DON, and FB_1_, with the highest rates amounting up to 100% (Figure 5a). For mycotoxins degradation, the *I. lacteus* and *C. subvermispora* MnPs behaved similarly, with transformation rates varying from 90.9–100% (for AFB_1_), 90.2–94.3% for ZEN, 34.1–43.6% for FB_1_, respectively. For DON, *Il*MnP1 had a relatively lower transformation rate of 76.8%, compared to 92.9–96.3% for *Il*MnP2, *Il*MnP4, *Il*MnP5, *Il*MnP6, and *Cs*MnP. *Pc*MnP1 and *Nf*MnP had similar transformation rates for DON and FB_1_ (42.9–43.5% and 22.2–30.2%, respectively). *Pc*MnP1 was less effective than *Cs*MnP in transforming AFB_1_ and ZEN (46.1% and 29.1% for *Pc*MnP1, and 78.4% and 42.2% for *Cs*MnP, respectively). However, both enzymes were outperformed by the *I. lacteus* and *N. frowardii* MnPs. Similar to AFB_1_ and ZEN reacted with MnPs, no large product peak was observed for DON incubated with MnPs in HPLC analysis. For DON and FB_1_, the reaction mixtures were subjected to UHPLC/MS-MS analysis but we were not able to identify a structurally determined product in the search, possibly because of the complexity of the reaction.

It is also noted that these MnPs are much diversified as shown in the phylogenic tree and their reciprocal amino acid sequence homology is as low as 48.0% (Figure 5b). This indicates that mycotoxin detoxification is a common feature shared by these MnPs and possibly other MnPs, as well.

### 2.4. Degradation of Mycotoxins Was Related to RB5 Decolorization

MnPs have been used in decolorization of synthetic dyes [27]. If the rate of dye decolorization by MnPs is positively related to that of mycotoxin degradation, dye decolorization could then be used as a safe indicator, speeding up the discovery of novel MnPs with mycotoxin-degrading ability or evolving MnPs with higher degrading ability. The structure of RB5 is given in Figure 1, which shares similar features (such as phenolic groups) with some mycotoxins but is much different by having the azo and sodium sulfonate groups. Using *Il*MnP5 and *Il*MnP6 as model MnPs, it was found that SOD slightly but significantly, while rutin largely inhibited the decolorization of a synthetic azo dye RB5, similar to that observed for AFB_1_ and ZEN detoxification (Appendix A). This demonstrated that radicals are also involved in RB5 decolorization. Thus, free radicals attacking could be overlappingly used by the MnPs in mycotoxin detoxification and RB5 decolorization. This suggests that RB5 decolorization may serve as an indicator of radical performance, and further, mycotoxin degradation. Indeed, the decolorization rates of RB5 by the eight MnPs were positively related to the degradation rates for the four major mycotoxins, with *R*^2^ equaling 0.710–0.945 (Figure 6a–d). Therefore, the ability to decolorize a synthetic dye (specifically RB5 as a prototype) could be used to estimate the potential of a MnP to degrade mycotoxins.

## 3. Discussion

Currently, there are approximately 500 known mycotoxins [28]. The structural diversity, in combination with co-contamination of multiple mycotoxins in food or feed, dwarfs the endeavors to use one or a limited number of enzymes for detoxification. However, in this study, we demonstrated that, MnPs can degrade at least four major mycotoxins instead of only one (i.e., AFB_1_, as have been reported previously) [11]. Although OTA has a phenolic hydroxyl group which is commonly the well-accepted substrate of MnPs, the tested MnPs were ineffective in its degradation. The underlying reason remains unknown.

For AFB_1_, degradation was mainly through the free radicals, which were generated by interaction of oxidized Mn^3+^ with the dicarboxylic acid malonate. AFB_1_-8,9-epoxide is labile in aqueous solution and its autonomous hydrolysis product can be monitored at 365 nm [29], the same wavelength used to detect disappearance of AFB_1_. Therefore, these together suggested that most chromophore groups (such as the benzene ring) in AFB_1_ had been eliminated in MnP treatment and the observed AFB_1_-8,9-epoxide (in UHPLC/MS-MS analysis) and new product peaks in HPLC analysis are likely minor parts of the products. The fact that AFB_1_-8,9-epoxide only constituted a small fraction of the products in our experiment is of significance. This is because AFB_1_-8,9-epoxide is not a desirable product and it irreversibly attaches to guanine residues to generate highly mutagenic DNA adducts [30].

The nature of other and major complex degradation products remains to be unveiled. This finding appears to be applicable to MnP-catalyzed ZEN degradation, as well. However, since the 20.3 min product peak was present, it could be imagined that direct reaction of MnP with ZEN could also happen. With these observed, there appeared to be some competition between the two pathways leading to ZEN degradation, i.e., direct reaction of a MnP with ZEN and indirect reaction via free radicals, since the degradation rates of ZEN by *Il*MnP5 and *Il*MnP6 in lactate buffer were higher than those in malonate buffer but lower than those in acetate buffer.

The huge number of mycotoxins prevents us from testing the efficacy of the eight MnPs in their degradation one by one. However, given the wide reactivity of free radicals generated by MnPs, amended by direct interaction of MnPs with substrates containing the phenolic hydroxy, phenylamine, and azino linkages [8], it is very likely that other mycotoxins can also be degraded. More importantly, it appears that the ability to degrade multiple mycotoxins is not restricted to one specific MnP but tends to be a common feature shared by this kind of enzymes. In this study, although the toxicity of degradation products of DON and FB_1_ was not evaluated, we observed much reduced toxicity (AFB_1_) or estrogenicity (ZEN) for AFB_1_ and ZEN degraded by *Il*MnP5 and *Il*MnP6 using the hydra and BLYES yeast assays, suggesting the application potential of these enzymes in mycotoxin detoxification. Nevertheless, it has to be admitted that these preliminary assays used much simplified model systems and cannot completely reflect the true scenarios where mycotoxins exert their roles in human and animals. For example, in addition to its toxicity, AFB_1_ is well-known for its carcinogenicity and causes cancer in the liver, kidney, and colon [31]. Therefore, a detailed study of the nature of the complex degradation products for all the tested mycotoxins, as well as their toxicity to cultured cells and animal models have to be systematically investigated. Two or more mycotoxins are commonly observed to co-exist but the types of mycotoxins vary among different foods and feeds [32]. If the MnP-catalyzed degradation products of certain specific combinations of mycotoxins are, in future, proved to be of less or even no toxicity to animals and human, MnP can truly serve as a candidate enzyme for detoxification because of its capability to degrade multiple mycotoxins.

We also demonstrated that RB5 decolorization may be used as a starting point to dictate discovery of new MnPs, through which the risk of handling mycotoxin will be greatly reduced. As MnP is classified in a large although expanding CAZy (Carbohydrate Active Enzymes) family (http://www.cazy.org/AA2.html) [33], it is therefore expected that the current finding of broad-spectrum mycotoxin-degrading enzymes will be used as a guiding framework to facilitate future discovery of robust mycotoxin-degrading enzymes with desirable properties. Additionally, this method can also be used for high-throughput screening of MnP variants with improved properties such as higher velocity of transformation, which may be generated from (semi)rational design or directed evolution. This is important since the time of incubation with the four mycotoxins is as long as 72 h in this study (Figure 5). From a practical perspective, a MnP with higher efficiency as well as higher velocity of transformation would be much more preferred.

One drawback of using MnPs for mycotoxins degradation (and potentially, detoxification) is the difficulty in obtaining large quantities of such enzymes. White-rot fungi, such as *P. chrysosporium*, *C. subvermispora*, and *I. lacteus*, are a natural microbial source rich in MnP genes. However, the white rots are divided into simultaneous and selective lignocellulose degraders, which differ in their way of regulating expression of the lignin-degrading enzymes (including MnPs) [14]. For instance, while *P. chrysosporium* gradually increases expression of MnPs (and other lignin-degrading enzymes) during culture on lignocellulose [15], the selective lignocellulose degraders *C. subvermispora* and *I. lacteus* only express MnPs at the early stage of culturing [12,34]. Therefore, care has to be taken in monitoring the expression of MnPs if such enzymes are to be produced from these microorganisms. MnPs can also be made in *Pichia pastoris* [35] or filamentous fungi [36]; however, they are notoriously hard to express recombinantly. In addition, the prosthetic group hemin has to be supplemented to the culture as an expensive additive for the formation of functional proteins. Thus future efforts can be placed in genetic engineering of the commonly used platform industrial microbes to facilitate large scale, heterologous production of MnPs.

Although MnPs were the only enzymes investigated in this study, because of the important role of free radicals involved, it is proposed that other oxidoreductases may also have the ability to degrade mycotoxins. Specifically, another lignin-degrading enzyme, laccase [37] (and probably other peroxidases as well), may also be used in mycotoxin degradation once a reaction condition is settled up favoring generation of mycotoxin-attacking radicals. Mediators (i.e., small molecules reacting with a laccase) play key roles in dictating the scope of a laccase and amplifying its catalyzing efficiency [38]. Both artificial and natural mediators can be screened if laccase is to be employed in mycotoxin degradation. Indeed, the Ery4 laccase from *Pleurotus eryngii* has been used to degrade AFB_1_, ZEN, FB_1_, and OTA in presence of a few structurally defined chemicals, such as acetosyringone and syringaldehyde, albeit with lower efficiency [31]. Interesting, the laccase/mediator system can degrade OTA but cannot act on DON, while the MnP/malonate system can degrade DON but not OTA. All in all, the versatility of manganese peroxidase makes it a candidate enzyme to simultaneously degrade multiple and much differing mycotoxins in food and feed, although a systematic investigation of the toxicity of the degradation products to animals and human has to be implemented ahead of that.

## 4. Conclusions

In summary, eight MnPs were found to be able to degrade four major mycotoxins, AFB_1_, ZEN, DON, and FB_1_ in the presence of a dicarboxylic acid malonate, in which free radicals play important roles. The ability to degrade multiple mycotoxins is not restricted to one MnP but tends to be shared by MnPs with much diversified sequences. We also demonstrated that RB5 decolorization may be used as a starting point to dictate discovery of new MnPs, through which the risk of handling mycotoxin will be greatly reduced. Our research suggests the possibility of using MnP as a candidate enzyme to degrade multiple mycotoxins simultaneously in food or feed if careful surveillance of the residual toxicity of degradation products is properly carried out.

## 5. Materials and Methods

### 5.1. Chemicals and Other Materials

AFB_1_, ZEN, DON, and the commercial *Nematoloma frowardii* manganese peroxidase (*Nf*MnP) were purchased from Sigma-Aldrich (St. Louis, MO, USA). FB_1_ and OTA were purchased from Pribolab (Beijing, China). Hemin was purchased from TCI (Tokyo, Japan). DNA polymerase, T4 ligase, and chromatographic grade reagents (acetonitrile, methanol, methanoic acid, acetic acid, and trifluoroacetic acid) were purchased from Thermo Fisher Scientific (Waltham, MA, USA). DNase I and TransScript One-Step gDNA Removal and cDNA Synthesis Supermix with oligo(dT) were purchased from TransGen (Beijing, China). Isopropyl-β-D-thiogalactoside (IPTG), SOD, DTT, and Rutin were purchased from Solarbio (Beijing, China). TRIZOL was from Invitrogen (Carlsbad, CA, USA). Lysozyme was purchased from Amresco (Solon, OH, USA). Ni-NTA agarose was purchased from QIAGEN (Duesseldorf, Germany). All other chemicals were of analytical grade or chromatographically pure, and were commercially available.

### 5.2. Plasmids, Bacterial Strains, and Cultural Conditions

The plasmids used in this study for expression of recombinant MnPs were pET-28a-*Il*MnP1, pET28a-*Il*MnP2, pET28a-*Il*MnP4, pET28a-*Il*MnP5, pET28a-*Il*MnP6, pET28a-*P*cMnP1, and pCold I-*Cs*MnP (Appendix A). *Irpex lacteus* CD2 was isolated from Shennong Nature Reserve (Hubei province, China) and maintained at 4 °C on potato-dextrose agar (PDA) plate. The *Escherichia coli* Trans1-T1 was used for gene cloning and plasmid propagation. The *E. coli* BL21 (DE3) strain was used for the expression of enzymes. These *E. coli* strains were cultivated at 37 °C with constant shaking at 220 rpm in Luria-Bertani (LB) broth medium: tryptone (10 g), yeast extract (5 g), and NaCl (10 g) in water (1 L, pH 7.0) containing appropriate antibiotics.

### 5.3. Construction of Recombinant Plasmids

The primers used in this study are listed in Appendix A. *I. lacteus* CD2 was grown for 5 d in the basal liquid medium. Total RNA was extracted using the TRIZOL reagent according to the manufacturer’s instructions and reverse transcribed to the first strand cDNA using the TransScript One-Step gDNA Removal and cDNA Synthesis Supermix with oligo (dT). The *Phanerochaete chrysosporium PcMnP1* (GenBank: J04980.1) [25] and *Ceriporiopsis subvermispora CsMnP* (GenBank: MG190336.1) [26] genes devoid of the sequences encoding the signal peptide were synthesized by BGI (Beijing, China). The three-dimensional structures of two homologs of *Pc*MnP1 and *Cs*MnP have been solved, which can be found at the protein structure database (http://www.rcsb.org/) with the entry numbers of 1MNP [39] (or 1YYD [40]) and 4CZN [41], respectively. *Pc*MnP1 has 79% amino acid sequence identity with 1MNP (or 1YYD) while *Cs*MnP has 84% identity with 4CZN. The *I. lacteus*, *P. chrysosporium*, and *C. subvermispora MnP* genes were amplified with gene specific primers using the following conditions: 95 °C for 2 min; then 30 cycles of 95 °C for 30 s, 54 °C for 30 s, and 72 °C for 1 min. The amplified genes of *IlMnP* enzymes and *PcMnP1* were restriction digested with *Eco*RI/*Not*I, gel purified, and ligated into the pET-28a(+) plasmid pre-digested with the same enzymes to obtain pET-28a-*Il*MnP1, pET28a-*Il*MnP2, pET28a-*Il*MnP4, pET28a-*Il*MnP5, pET28a-*Il*MnP6, and pET28a-*Pc*MnP1 (Appendix A). The PCR product of *CsMnP* was digested with *Nde*I/*Bam*HI, gel purified, and ligated into the pre-digested pCold I (digested with *Nde*I/*Bam*HI) to obtain pCold I-*Cs*MnP. The seven recombinant plasmids were transformed into *E. coli* BL21(DE3) competent cells for gene expression.

### 5.4. Expression of Manganese Peroxidases

The *E. coli* BL21(DE3) strains harboring pET-28a-*Il*MnP1, pET-28a-*Il*MnP2, pET-28a-*Il*MnP4, pET-28a-*Il*MnP5, pET-28a-*Il*MnP6, or pET-28a-*Pc*MnP1 were cultured in LB medium supplemented with 50 μg/mL of kanamycin at 37 °C overnight with shaking at 220 rpm. These pre-cultures were individually inoculated into 200 mL LB medium. The culture was continued at 37 °C for approximately 2 h. When the optical density at 600 nm (OD_600_) reached 0.6–0.8, IPTG was added to a final concentration of 1 mM for induction of MnPs expression [13]. For the *E. coli* BL21(DE3) strain containing pCold I-*Cs*MnP, when OD_600_ reached 0.5, the bacterium was rapidly chilled to 10 °C by soaking the culture flask in a water-ice bath, followed by shaking at 220 rpm at 10 °C for 30 min. IPTG was added to a final concentration of 0.2 mM and the cells were grown at 10 °C with shaking at 220 rpm for 24 h. Six hour after IPTG was added, 1 M CaCl_2_ and 10 g/L hemin were supplied continuously for another 9 h at rates of 22 μL/h and 220 μL/h to the 200 mL LB culture medium, respectively. After induction, the cells were harvested by centrifugation at 12,000 *g* for 2 min [26].

### 5.5. Refolding and Purification of the Recombinant MnPs

For *E. coli* expressing recombinant MnPs from *I. lacteus* and *P. chrysosporium* MnPs, the harvested cell pellets were re-suspended in 50 mM Tris-HCl, 10 mM EDTA, and 5 mM DTT (pH 8.0). Lysozyme was added to a final concentration of 2 mg/mL and the cells were incubated on ice for 1 h. Then, 20 μL of DNase I was added and the incubation was continued on ice for 30 min. Subsequently, the cells were centrifuged at 12,000 *g* for 30 min at 4 °C. The cell debris was washed twice with 20 mM Tris-HCl, 1 mM EDTA, and 5 mM DTT (pH 8.0), followed by incubation in 50 mM Tris-HCl, 8 M urea, 1 mM EDTA, and 1 mM DTT (pH 8.0) on ice for 1 h. To optimize the parameters for recovery of active enzyme from the inclusion bodies, the refolding was carried out in different conditions. The parameters including concentrations of urea, GSSG, and hemin and pH were investigated, while the concentrations of enzyme, EDTA, and DTT were kept constant during the refolding. The efficiency of refolding was indicated by the MnP activity. Refolding of the MnPs were conducted under the optimized conditions (pH 9.5, 50 mM Tris-HCl buffer, 0.6 M urea, 0.5 mM GSSG, 0.1 mM DTT, 10 μM hemin, 5 mM CaCl_2_, 0.1 mg/mL protein for *Il*MnP1; pH 9.5 50 mM Tris-HCl buffer, 0.5 M urea, 0.7 mM GSSG, 0.1 mM DTT, 10 μM hemin, 5 mM CaCl_2_, 0.1 mg/mL protein for *Il*MnP2, *Il*MnP5 and *Il*MnP6; pH 9.5 50 mM Tris-HCl buffer, 0.5 M urea, 0.7 mM GSSG, 0.1 mM DTT, 10 μM hemin, 5 mM CaCl_2_, 0.1 mg/mL protein for *Il*MnP4; pH 9.5 50 mM Tris-HCl buffer, 1 M urea, 0.4 mM GSSG, 0.1 mM DTT, 10 μM hemin, 5 mM CaCl_2_, 0.1 mg/mL protein for *Pc*MnP1) for 10 h at 15 °C. After refolding, the crude enzymes were centrifuged at 12,000 *g* for 10 min at 4 °C and the insoluble fractions were discarded. The supernatants containing the refolded MnP were concentrated through a 10 kDa cut-of centrifuge filter, followed by dialysis against buffers of different pHs (pH 6.0, 20 mM Na_2_HPO_4_-citric acid buffer for *I*lMnP1; pH 5.0 20 mM HAc-NaAc buffer for *Il*MnP2, *Il*MnP5, *Il*MnP6; pH 6.5 20 mM Na_2_HPO_4_-citric acid buffer for *Il*MnP4 and *Pc*MnP1). The crude enzymes were further purified by a HiTrap Q HP anion exchange column (GE Health, Fairfeld, CT) pre-equilibrated with the same buffer. The proteins were eluted with a linear gradient of 0–1.0 M NaCl, and fractions containing pure and active enzymes were pooled [13].

The *E. coli* cell pellet expressing recombinant *Cs*MnP was harvested from 100 mL of culture medium, re-suspended in 10 mL of 50 mM Tris-HCl buffer (pH 8.0) containing 1 mM CaCl_2_, and homogenized by sonication. After centrifugation at 12,000 *g* for 30 min, the cell debris was discarded and the supernatant was collected, which was passed through a Ni affinity column resin. The resin was washed with 50 mM Na_2_HPO_4_-NaH_2_PO_4_ buffer (pH 7.5) containing 500 mM NaCl, 1 mM CaCl_2_, and 20 mM imidazole to remove the nonspecifically bound proteins. The bound MnPs were eluted with 50 mM Tris-HCl (pH 7.5) containing 500 mM NaCl, 1 mM CaCl_2_, and 40/60/80/100/200/500 mM imidazole, respectively. The eluents were analyzed for purity by SDS-PAGE. The purified MnPs were stored in 50 mM Tris-HCl (pH 7.5) at 4 °C containing 1 mM CaCl_2_ until used [26]. The purified recombinant manganese peroxidases were analyzed on SDS-PAGE (Appendix A).

### 5.6. Measurement of MnP Activity

The MnP activity was measured by monitoring the oxidation of 2,2′-Azino-*bis*(3-ethylbenzothiazoline-6-sulphonic acid (ABTS, ε420 = 36,000 M^−1^·cm^−1^) at 420 nm, in a buffer containing 50 mM malonate, 1 mM ABTS, 1 mM MnSO_4_, and 0.1 mM H_2_O_2_ (pH 5.0 and 25 °C) [13]. One unit (1 U) of MnP activity was defined as the amount of enzyme that produced 1 μmol of product per minute under the standard assay condition.

### 5.7. Mycotoxin Degradation

The activities of the enzymes were first determined using ABTS as the substrate. Each of the calibrated enzymes (0.5 U/mL each) was then incubated with mycotoxins (AFB_1_ and ZEN 5 μg/mL; DON, and FB_1_, 10 μg/mL; OTA, 50 μg/mL) in 70 mM malonate buffer supplemented with 1 mM MnSO_4_ and 0.1 mM H_2_O_2_. The reaction was carried out at 30 °C. Periodically, samples were taken out and three volumes of methanol were added to terminate the reaction. Each reaction was repeated three times.

### 5.8. RB5 Decolorization

The reactions were carried out at 30 °C in a total volume of 200 μL containing 50 mM malonate buffer (pH 5.0), 0.1 mM H_2_O_2_, 1 mM Mn^2+^, 0.25 U/mL each of the MnPs, and 50 μg/mL of dye. During the incubation, the color change was detected by measuring the OD_556_ (RB5). The rate of decolorization was calculated using the following formula: decolorization (%) = [(Ai−At)/Ai] × 100, where Ai and At are the absorbance at the initial and a given time [13].

### 5.9. BLYES Assay

The *Saccharomyces cerevisiae* BLYES strain was inoculated into 30 mL YMM ((NH_4_)_2_SO_4_) 1.7 g/L, CuSO_4_ 12 mg/L, FeSO_4_ 684 µg/L, KH_2_PO_4_ 11.6 g/L, KOH 3.6 g/L, MgSO_4_ 171 µg/L, D-(+)-glucose 20 g/L, biotin 20 µg/L, pantothenic acid 400 µg/L, inositol 1 mg/L, pyridoxine 400 µg/L, thiamine 400 µg/L, adenine 42.7 mg/L, arginine HCl 17.1 mg/L, aspartic acid 100 mg/L, glutamic acid 85.5 mg/L, histidine 42.73 mg/L, isoleucine 25.64 mg/L, lysine HCl 25.64 mg/L, methionine 17.1 mg/L, phenylalanine 21.4 mg/L, serine 320.4 mg/L, threonine 192 mg/L, tyrosine 25.7 mg/L) in a baked 250 mL glass flask. The cells were cultured at 28 °C with constant shaking at 200 rpm to an OD_600_ of 0.6. ZEN was dissolved in 70 mM malonate buffer to 5 μg/mL supplemented with 1 mM MnSO_4_ and 0.1 mM H_2_O_2_. Twenty microliter of appropriately diluted ZEN pre-treated or non-treated with *Il*MnP5 or *Il*MnP6 were mixed with 200 µL BLYES and the estrogenicity was checked by measuring the bioluminescence of the cells collected 6-h post treatment for analysis [20,21].

### 5.10. Hydra Assay

The hydra was maintained clean and free from bacteria and fungi contamination by treating with diluted iodine solution (2.7 ppm) periodically. The assay was performed by exposing the hydra to AFB_1_ treated or nontreated with an MnP. Fifty μg/mL of AFB_1_ were incubated with *Il*MnP5 or *Il*MnP6 (0.5 U/mL each) in 70 mM malonate buffer supplemented with 1 mM MnSO_4_ and 0.1 mM H_2_O_2_. The reaction was carried out at 30 °C for 10 h. Each test dish contained 1 mL of test solution and three normal healthy hydra. The hydra were examined for signs of toxicity at 20 h and 40 h, respectively. The toxic endpoint was determined by the “tulip” or “disintegration” stage of the hydra. In each test, experimental treatments were compared with untreated and solvent controls [19].

### 5.11. HPLC and LC-MS/MS Analyses

HPLC analysis of AFB_1_, ZEN, DON, and OTA was performed using a SHIMADZU 20A series instrument (Kyoto, Japan) with an Agilent ZORBAX SB-C18 column (5 µm, 4.6 mm × 250 mm) (Santa Clara, CA, USA). The elution condition for AFB_1_ and ZEN was set as: no acetonitrile (ACN), 4 min; 0–100% ACN, 25 min; 100% ACN, 6 min, at a flow rate of 0.8 mL/min. AFB_1_ and ZEN were monitored at 365 nm or 316 nm [42], respectively. The elution condition for DON was set as: 20% methanol, 20 min; 20–100% methanol, 1 min; 100% methanol, 6 min, at a flow rate of 0.8 mL/min. DON was monitored at 220 nm. The mobile phase for OTA was mixed CAN:H_2_O:HAc (99:99:2) and the flow rate was 1.0 mL/min. OTA was monitored for its absorbance at 333 nm.

AFB_1_, ZEN transformation products were analyzed by using LC-MS/MS, which was carried out by coupling a SHIMADZU Nexera UHPLC system (Kyoto, Japan) to an AB-SCIEX 5600+ Triple TOF mass spectrometer (Waltham, Massachusetts, USA). The solvent A for LC is mixed ACN:methanol (1:1) and solvent B is 0.1% fomic acid. The program was set as: 30–70% solvent A, 10 min; 70% solvent A, 8 min; 100% solvent A, 2 min; 30% solvent A, 5 min. The parameters for MS analysis were: positive and high-sensitivity mode; GS1, 50 psi; GS2, 50 psi; curtain gas, 25 psi; temperature, 500 °C; ion spray voltage floating, 5, 500 V; CE energy, 35 V ± 15 V. Degradation of FB_1_ was analyzed by using UHPLC-MS/MS, which was carried out by the same method as above.

### 5.12. Phylogenetic Analysis

The alignment of the amino acid sequences of 8 MnPs were conducted using the clustalW algorithm of MEGA-X. Phylogenetic tree constructions were performed using the neighbor-joining method of MEGA-X. The reliability of the trees was tested by bootstrap analysis and the parameter was set to 1000. The other parameters used default values.

## Figures and Tables

**Figure 1 toxins-11-00566-f001:**
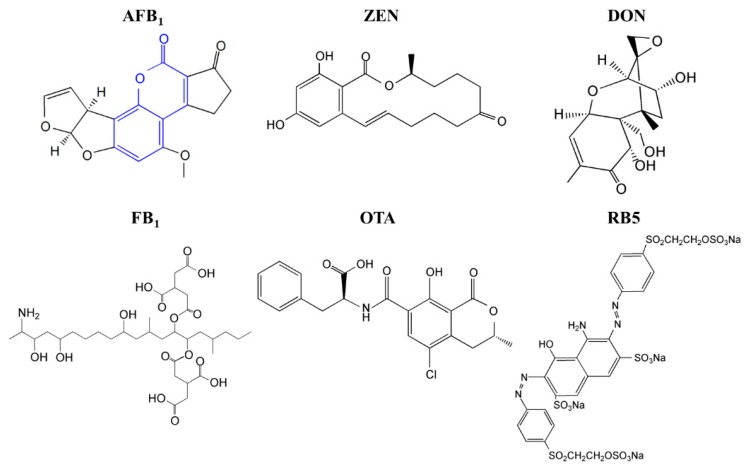
The chemical structures of five major mycotoxins (aflatoxin B_1_ (AFB_1_), zearalenone (ZEN), deoxynivalenol (DON) fumonisin B_1_ (FB_1_), ocharatoxin A (OTA)) and the synthetic dye RB5. The highlighted part in blue is a coumarin structure, which is a derivative of the lignin monomer *p*-coumaryl alcohol.

**Figure 2 toxins-11-00566-f002:**
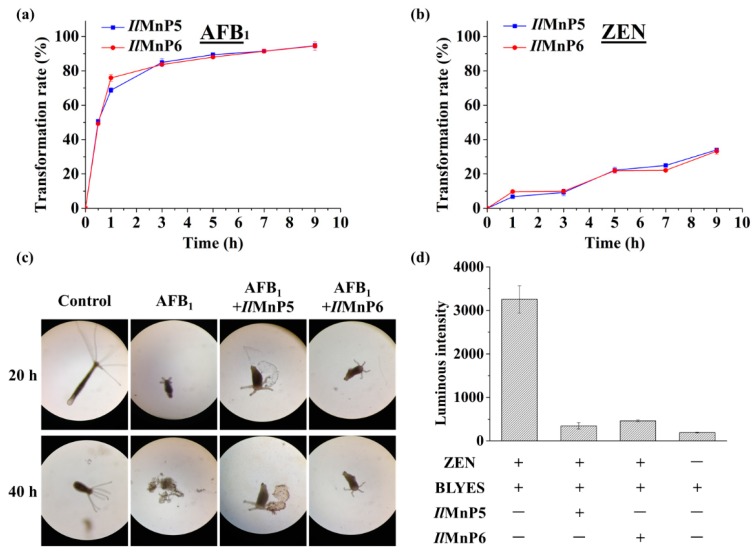
Degradation of AFB_1_ and ZEN by *Il*MnP5 and *Il*MnP6 in a malonate buffer. (**a**,**b**) Time-course analysis of AFB_1_ (**a**) and ZEN (**b**) degradation by *Il*MnP5 and *Il*MnP6. (**c**) Treating AFB_1_ using *Il*MnP5 and *Il*MnP6 alleviated its toxicity on hydra. (**d**) *Il*MnP5 and *Il*MnP6 reduced the estrogenic activity of ZEN to the BLYES yeast.

**Figure 3 toxins-11-00566-f003:**
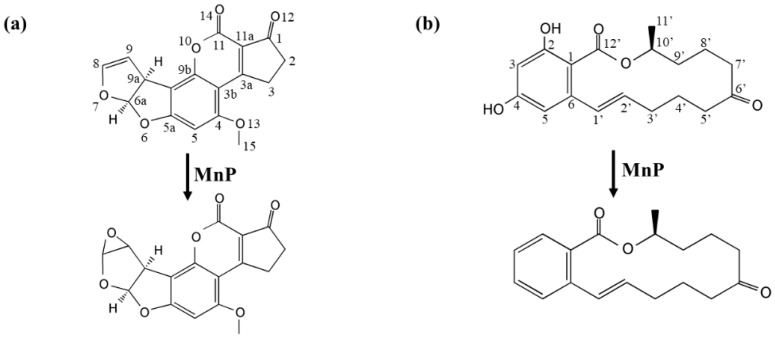
Identified reaction products with proposed structures in AFB_1_ (**a**) and ZEN (**b**) treated with *Il*MnP5 and *Il*MnP6, which were deduced from LC-MS/MS and literature review.

**Figure 4 toxins-11-00566-f004:**
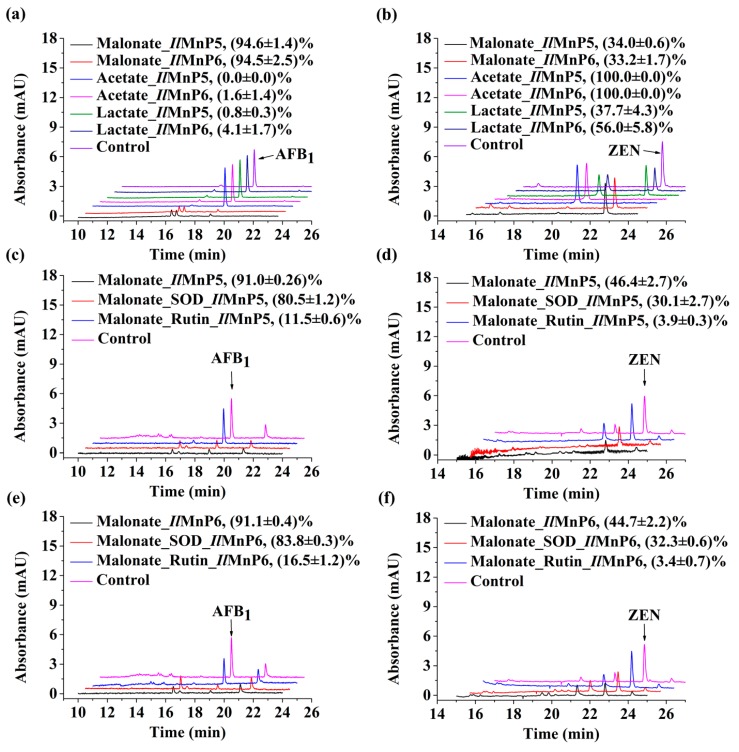
Radicals play an important role in mycotoxin degradation by *Il*MnP5 and *Il*MnP6. (**a**,**b**) Degradation of AFB_1_ (**a**) and ZEN (**b**) in malonate, lactate, or acetate buffers. (**c**,**d**) Effects of SOD and rutin on the degradation of AFB_1_ (**c**) and ZEN (**d**) by *Il*MnP5. (**e**,**f**) Effects of SOD and rutin on degradation of AFB_1_ (**e**) and ZEN (**f**) by *Il*MnP6. Note that the height of the traces of ZEN degradation was doubled for ease of observation. The degradation percentage was labeled for each enzyme-treated sample, represented as average ± standard deviation.

**Figure 5 toxins-11-00566-f005:**
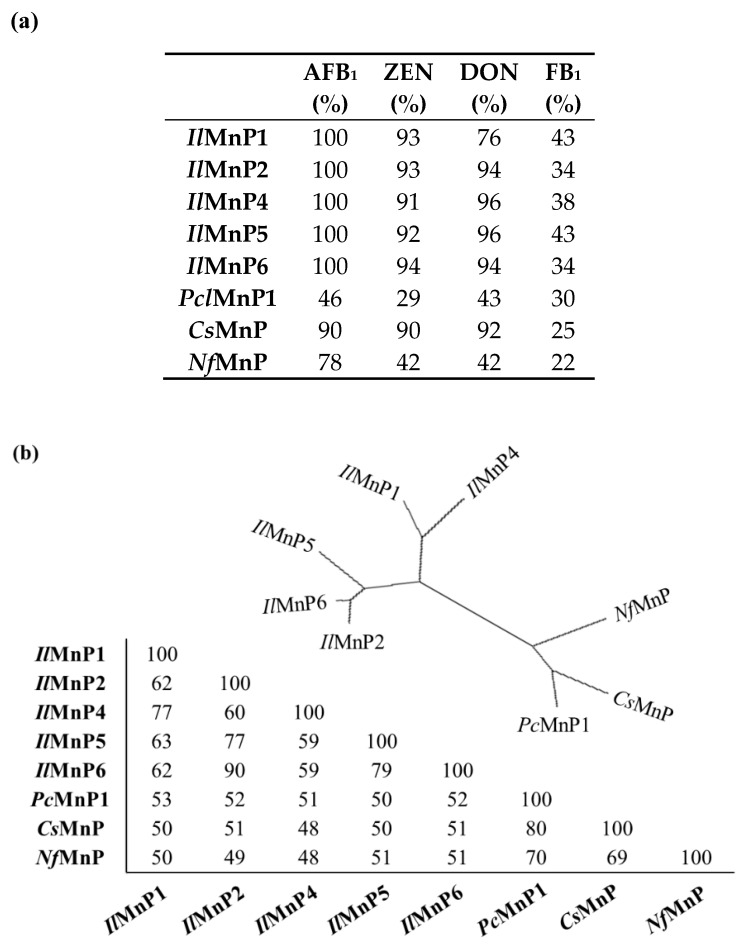
Degradation of multiple mycotoxins is a feature shared by eight MnPs. (**a**) Degradation percentages for five main mycotoxins by eight MnPs from *I. lacteu*, *P. chrysosporium*, *C. subvermispora*, and *N. frowardii*. The mycotoxins were incubated for 72 h with one of the enzymes (0.5 U/mL each) at 30 °C. (**b**) Phylogenic tree and the reciprocal amino acid sequence identity of the eight MnPs.

**Figure 6 toxins-11-00566-f006:**
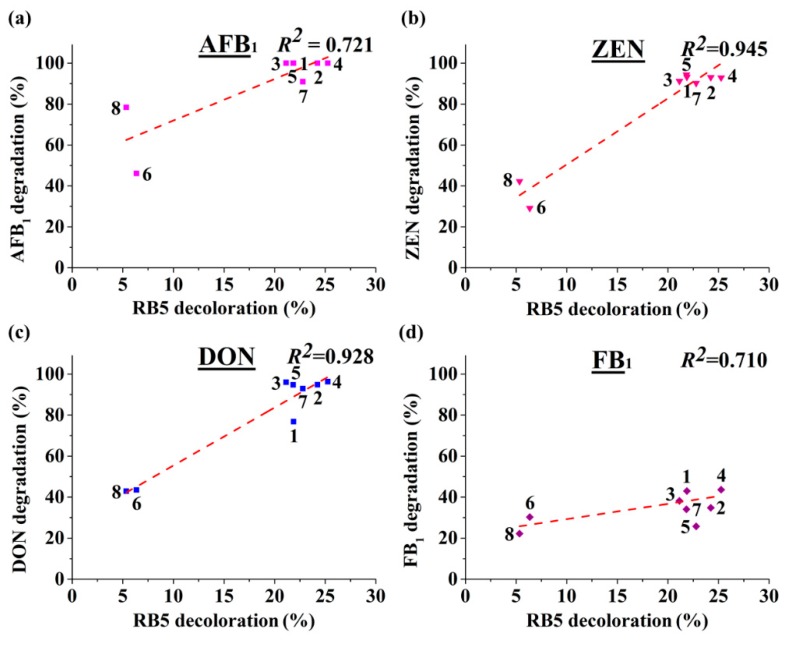
Degradation of four major mycotoxins (AFB_1_, ZEN, DON, and FB_1_, 72 h) was positively related to RB5 decolorization (9 h). The enzymes (*Il*MnP1, *Il*MnP2, *Il*MnP4, *Il*MnP5, *Il*MnP6, *Pc*MnP1, *Cs*MnP, and *Nf*MnP) are labeled with the numbers 1–8, respectively. (**a**) Degradation percentages of AFB_1_ was positively related to RB5 decolorizations by enzymes (*R*^2^ = 0.721). (**b**) Degradation percentages of ZEN was positively related to RB5 decolorizations by enzymes (*R*^2^ = 0.945). (**c**) Degradation percentages of DON was positively related to RB5 decolorizations by enzymes (*R*^2^ = 0.928). (**d**) Degradation percentages of FB_1_ was positively related to RB5 decolorizations by enzymes (*R*^2^ = 0.720).

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
