# Peer review of "Degradation of Four Major Mycotoxins by Eight Manganese Peroxidases in Presence of a Dicarboxylic Acid"

_toxins, 2019, doi:10.3390/toxins11100566_

Round 1
Reviewer 1 Report
The authors report the ability of MnP to not only degrade aflatoxin B1, but also a handful of other major mycotoxins that are major health concerns. The results of this paper show that MnP enzymes have a broad substrate specificity and utilise the generation or free radicals to transform these compounds. The paper is generally well written and appropriate experiments conducted.
It is unclear why IlMnP5 and IlMnP6 were chosen as the initial point of this study. Maybe this could be made clear in the introduction. Have these two enzymes been shown to degrade AFB1 in previous studies. What is the definition of a unit of enzyme? This was not obvious in the methods, should be section 5.6. Why were hydra used for the toxicity study? These are more commonly used for metal toxicity. A brief description of why these have been used and previous studies that demonstrate their utility for showing the toxicity of mycotoxins. Is there a better way of numerically demonstrating the survival/fitness of the hydra? The figure 2c is good but not very informative. Were any replicates conducted? The LCMS analysis of AFB1, paragraph should start for the sentence starting line 109. Figure s4a does not show the MS spectra with the parent ion. The MS/MS does not appear to have much more signal than above the noise. Also, more detail of the results is written in the discussion, this should be moved to the results section to describe the results observed. The discussion section could then be simplified to be more concise. Figure 5a, it is unclear how long these have been incubated 4. Or how much enzyme was used, was this 0.5 units/ml? Section 2.4 could be strengthened by elaborating on the first sentence to raise the question whether the decolourization activity could be used as a screening tool to quickly assess if an MnP could be used degradation. Line 217. It should be stated that AFB1-8,9-epoxide is not a desirable product as this is the compound that is known to chelate DNA. Some clarity in this part of the discussion is required. Paragraph starting 251. Could be give more strength by using the dye as a high throughput screen for not only identifying new enzymes but also in the evolution of known enzymes to improve activity and solubility. Making the enzymes more soluble is going to decrease the production costs and improve the commercial prospects of this family of enzymes. The enzymes described do not degrade the mycotoxins quickly, this should be discussed. Again, this activity may also be improved, or is this an issue with enzymes that lack substrate specificity.
Minor points
Avoid colloquial language (used extensively through the text)
Line 6 and 46 – “Attractive means” Line 30, “almost 20 thousand” would be better than “19, 757” Line 31, is there a better word for feed raw material samples? Line32, reintroduce mycotoxins. Line 39, “agronomic means have proven..” Line 50, “This requires that enzymes must have a wide substrate specificity on multiple mycotoxins”. Line 85. These are percent degraded rather than rates. A rate would be how much µmole substrate is degraded by a µmole of enzyme per minute. Line 105. “which is consistent with” Throughout – hydra not hydras Line 309. Describe the plasmids used in the study. It is only two, and much easier to describe than look at the vector maps in the supplementary materials. Section 5.3 it is not clear hwo the pEt28a vectors were made, what were the cloning sites?
Reviewer 2 Report
Dear Authors,
Your manuscript entitled "Degradation of Four Major Mycotoxins by Eight Manganese Peroxidases in Presence of a Dicarboxylic Acid" is of quality and proposes a new alternative to mycotoxin degradation.
The article is clear, original and research design is appropriate. Nevertheless, I would like to suggest minor and major remarks in order to improve your scientific contribution.
In general, the manuscript needs to be meticulously reviewed in order to correct minor spell and writing errors. For instance, line 139, the abbreviation of SOD is incomplete, the same error is observed in lines 188-190. Another example can be founded in lines 140-142 where a part of the phrase is dismissed. Figure 2 needs to be improved in size and quality to appreciate the effect of the different MnPs on hydras.
The introduction and discussion section could be improved with more and more recent bibliography, especially in mycotoxin section, as well as a it would be appreciated a quick description of the different white-rot fungi strains used in this study.
Other minor corrections concern graphical data such as Figure 4, in which absorbance values are difficult to estimate due to the type of figure.
A statistical analysis needs to be performed in order to validate the differences between treatments and results need to be shown in graphical data.
Concerning results, it is well-know that several mycotoxins can be degraded in other toxic compounds such is the case of AFB1. Please provide information about the research of other toxic compounds in your different analyses.
Does dicarboxylic acid malonate is demonstrated as a toxic compound? Please provide information about it.
Are control analyses with commercial MnPs standards were performed in this study? Several control tests are dismissed. I would like to suggest the use of a commercial MnP standard in order to verify the RB5 decolorization proposal.
The use of white-rot fungi MnPs as degrading compounds of mycotoxins tested in mycotoxin derivates such as 3-epi-deoxynivalenol and deoxynivalenol-3-glucoside may also be of interest.
Reviewer 3 Report
The article presented on the degradation of mycotoxins by manganese peroxidases has an interest in expanding knowledge to a greater number of mycotoxins, and the article deserves to be published in Toxins.
However, any reference to the RB5 sulfonic azocolorant should be removed from the article, as it does not provide data of interest regarding human or animal feed.
On the other hand, in Figures 5a and 5b, decimal numbers must be eliminated because they lack statistical significance.

Round 2
Reviewer 3 Report
With the corrections made, the article can be published in its new version.